# L-MSA: Layer-wise Fine-tuning using the Method of Successive Approximations

## Abstract

With the emergence of large-scale models, the machine learning community has witnessed remarkable advancements. However, the substantial memory consumption associated with these models has emerged as a significant obstacle to large-scale training. To mitigate this challenge, an increasing emphasis has been placed on parameter-efficient fine-tuning methodologies, which adapt pre-trained models by fine-tuning only a subset of parameters. We observe that in various scenarios, fine-tuning different layers could lead to varying performance outcomes, and selectively fine-tuning certain layers has the potential to yield favorable performance results. Drawing upon this insight, we propose L-MSA, a novel layer-wise fine-tuning approach that integrates two key components: a metric for layer selection and an algorithm for optimizing the fine-tuning of the selected layers. By leveraging the principles of the Method of Successive Approximations, our method enhances model performance by targeting specific layers based on their unique characteristics and fine-tuning them efficiently. We also provide a theoretical analysis within deep linear networks, establishing a strong foundation for our layer selection criterion. Empirical evaluations across various datasets demonstrate that L-MSA identifies layers that yield superior training outcomes and fine-tunes them efficiently, consistently outperforming existing layer-wise fine-tuning methods.

## 1 Introduction

With the increasing application of large-scale models across diverse task domains(Devlin et al., 2019; Dosovitskiy et al., 2021), domain-specific fine-tuning has emerged as a pivotal strategy to bolster their effectiveness in downstream tasks(Käding et al., 2017; Raffel et al., 2020). However, these fine-tuning methods are often resource-intensive, presenting significant challenges in the development of large-scale models. Efforts to address these challenges have led to the development of Parameter-Efficient Fine-Tuning (PEFT) techniques, which aim to mitigate computational costs. These techniques encompass various approaches, such as prompt-based methods(Diao et al., 2022; Hambardzumyan et al., 2021; Lester et al., 2021; Liu et al., 2023), adapter methods(Diao et al., 2023; Houlsby et al., 2019; Hu et al., 2021), and selective methods(Li et al., 2024; Liu et al., 2021; Zaken et al., 2021).

Among the array of Parameter-Efficient Fine-Tuning (PEFT) techniques, layer-wise fine-tuning algorithms have emerged as a promising solution(Lee et al., 2022; Pan et al., 2024). Instead of updating all parameters simultaneously, layer-wise fine-tuning focuses on iteratively fine-tuning individual layers of the model. This approach not only reduces computational costs but also allows for more targeted adjustments, potentially leading to improved performance on downstream tasks.

However, the specific layer to fine-tune may vary based on the relationship between the source and target datasets. To explore this, we conduct experiments with a Data-efficient Image Transformer (DeiT)-Tiny (Touvron et al., 2021) in three scenarios:

1. Pre-training on ImageNet(Deng et al., 2009a) and fine-tuning on CIFAR-100(Krizhevsky, 2009).

2. Pre-training on CIFAR-100 and transforming the input data by element-wise multiplication with a matrix, where each entry is $e^x$ and $x$ follows a standard normal distribution. Fine-tuning is then performed on the transformed data.

3. Generating two sets of random labels for the CIFAR-100 inputs, pre-training on one set of the labels, and fine-tuning on the other.

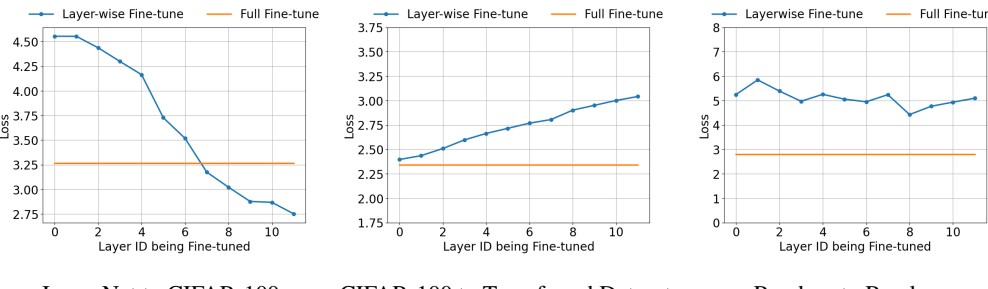

| ImageNet to CIFAR-100 | CIFAR-100 to Transferred Dataset | Random to Random |

Figure 1: Layer-wise Fine-tuning in Different Scenarios

In each case, we visualize the outcomes of layer-wise fine-tuning compared to full fine-tuning after a single epoch, with consistent observations even over extended training. In case 1, where the dataset shares similar low-level features but different high-level features compared to the original data, fine-tuning later layers outperforms earlier layers. Conversely, in case 2, with similar high-level features but different low-level features, fine-tuning earlier layers yields better performance. Finally, in case 3 involving random tasks, all layers are equally significant, and fine-tuning individual layers alone may not suffice. This variability raises the question of whether we can algorithmically determine which layer(s) to fine-tune and how to perform effective layer-wise fine-tuning.

To address the aforementioned challenge, we propose L-MSA, a novel layer-wise fine-tuning approach that consists of two core components: a metric for layer selection and an algorithm for optimizing the fine-tuning of the selected layer. This targeted optimization seeks to enhance overall model performance by leveraging the specific strengths of different layers.

We leverage the principles of the Method of Successive Approximations (MSA) (Chernousko & Lyubushin, 1982; Li et al., 2018) within our L-MSA framework, addressing both layer selection and layer fine-tuning. The first component of our approach introduces a novel metric, derived from the state and co-state variables in MSA, which serves as the criterion for selecting layers. The second component focuses on utilizing the MSA to optimize the fine-tuning of the selected layers. This integrated approach ensures efficient optimization by systematically refining the layer-wise fine-tuning process, ultimately leading to improved performance.

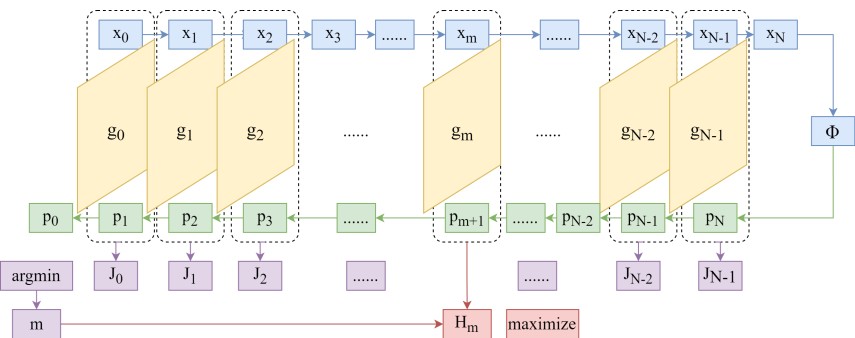

Figure 2: Overview of our proposed L-MSA method: We begin with a feed-forward pass to compute the state $x_i$ at each layer, followed by a back-propagation step to determine the co-state $p_i$. Utilizing both $x_i$ and $p_i$, we compute a metric $\hat{J}_n$ for every layer, as defined in equation 8, to guide layer selection. We then select the layer with the smallest metric, denoting its index as $m$, and maximize $H_m$ over the parameters $\theta_m$. By fixing $m$ and repeating these steps iteratively, we refine the layer parameters, converging toward a desired solution that enhances model performance.

Furthermore, we provide a comprehensive theoretical analysis of our L-MSA approach within the context of deep linear networks(Arora et al., 2018b). This analysis clarifies the metric we utilize for

optimal layer selection, framed within a greedy one-step optimization framework. By establishing a solid theoretical foundation, we pave the way for understanding how our method effectively enhances model training.

Finally, we evaluate our approach across various datasets and tasks, utilizing multiple model architectures, and compare L-MSA with baseline layer-wise fine-tuning methods. Our results demonstrate that the metric proposed in L-MSA effectively identifies the layers that will yield better training outcomes. L-MSA consistently outperforms most baselines, achieving top performance in several tasks and ranking as the most effective method overall, reinforcing the practical applicability of our approach in real-world scenarios. We also conduct ablation tests, highlighting the metric's effectiveness in layer selection and the advantages of using MSA to optimize the chosen layers.

We summarize our key contributions as follows:

- We experimentally show that in various scenarios, fine-tuning different layers could lead to varying performance outcomes, and selectively fine-tuning certain layers has the potential to yield favorable performance results.
- We propose the L-MSA method, which introduces a new criterion for selecting layers to fine-tune, and we also propose utilizing the method of successive approximations for layer-wise fine-tuning within our L-MSA approach, ensuring efficient optimization and improved learning outcomes.
- We provide a theoretical analysis of our approach in the context of deep linear networks, clarifying the metric for optimal layer selection within a greedy one-step framework.
- We empirically validate the effectiveness of our methodology in accurately identifying and efficiently fine-tuning the target layer across diverse datasets.

## 2 L-MSA: LAYER-WISE FINE-TUNING USING THE METHOD OF SUCCESSIVE APPROXIMATION

Adopting the control viewpoint for layer-wise fine-tuning offers a structured optimization process through Pontryagin's Maximum Principle (PMP)(Pontryagin et al., 1962). This perspective treats each layer as part of a controlled dynamical system, enabling precise adjustments to specific layers by assessing their impact on the overall loss via the Hamiltonian. Consequently, this method facilitates efficient fine-tuning by focusing on layers that offer the most significant performance improvement, thereby making the optimization process more systematic and effective.

### 2.1 BACKGROUND: PONTRYAGIN'S MAXIMUM PRINCIPLE AND METHOD OF SUCCESSIVE APPROXIMATION

In supervised learning, given a collection of $K$ sample input-label pairs $\{x^i, y^i\}_{i=1}^K$, our objective is to infer and approximate a function $F : \mathcal{X} \to \mathcal{Y}$ that accurately maps input data instances $x^i$ to their corresponding target outputs $y^i$. To view supervised learning within the dynamical systems framework, particularly relevant to deep and residual architectures, we consider the inputs $x = (x^1, x^2, \cdots, x^K) \in \mathbb{R}^{d \times K}$ as the initial condition of a system of ordinary equations

$$\dot{x}_t^i = f\left(t, x_t^i, \theta_t\right), \quad x_0^i = x^i, \quad 0 \le t \le T, \quad i = 1, \ldots, K, \tag{1}$$

where $\theta : [0, T] \to \Theta$ is the control parameters and $x_t = (x_t^1, \cdots, x_t^K) \in \mathbb{R}^{d \times K}$. In this context, $f(t, x_t^i, \theta_t)$ encapsulates the transformation process within the neural network, while $\theta_t$ represents the parameters at time $t$ that govern this transformation.

The supervised learning problem can be formulated as

$$\min_{\theta \in \mathcal{U}} \sum_{i=1}^K \Phi_i\left(x_T^i\right) + \int_0^\top L\left(\theta_t\right) dt, \tag{2}$$

$$\dot{x}_t^i = f\left(t, x_t^i, \theta_t\right), \quad x_0^i = x^i, \quad 0 \le t \le T, \quad i = 1, \ldots, K,$$

where $\Phi_i(\cdot) := \phi(\cdot, y^i)$ is the loss function, and $L : \Theta \to \mathbb{R}$ is a running cost, or the regularization term.

We define the Hamiltonian $H \colon [0, T] \times \mathbb{R}^d \times \mathbb{R}^d \times \Theta$ given by

$$H(t, x, p, \theta) = p \cdot f(t, x, \theta) - L(\theta) \tag{3}$$

Pontryagin's Maximum Principle(PMP)(Pontryagin et al., 1962) shows a set of necessary conditions for optimal solutions to equation 2, which provides an alternative numerical algorithm to train equation 2 and its discrete-time formulation.

**Theorem 2.1 (Pontryagin's Maximum Principle)** *Let $\theta^* \in \mathcal{U}$ be an essentially bounded optimal control, i.e. a solution to equation 2 with $ess\ sup_{t \in [0,T]} \|\theta_t^*\|_\infty < \infty$ (ess sup denotes the essential supremum). Denote by $x^*$ the corresponding optimally controlled state process. Then, there exists an absolutely continuous co-state process $P^* \colon [0, T] \to \mathbb{R}^d$ such that the Hamilton's equations*

$$\dot{x}_t^* = \nabla_p H\left(t, x_t^*, P_t^*, \theta_t^*\right), \qquad x_0^* = x,$$
$$\dot{P}_t^* = -\nabla_x H\left(t, x_t^*, P_t^*, \theta_t^*\right), \quad P_T^* = -\nabla\Phi\left(x_T^*\right), \tag{4}$$

*are satisfied. Moreover, for each $t \in [0, T]$, we have the Hamiltonian maximization condition*

$$H\left(t, x_t^*, P_t^*, \theta_t^*\right) \geq H\left(t, x_t^*, P_t^*, \theta\right) \text{ for all } \theta \in \Theta. \tag{5}$$

Consider an $N$-layer deep neural network, which can be interpreted as a discrete-time formulation of equation 2. Within this framework, the supervised learning problem can be expressed as follows:

$$\min \sum_{i=1}^{K} \Phi_i\left(x_N^i\right) + \sum_{n=0}^{N-1} \delta_t L\left(\theta_n\right)$$
$$x_{n+1}^i = g_n(x_n^i, \theta_n), \; x_0^i = x^i, \; i = 0, 1, \cdots, K. \tag{6}$$

Here $g_n(x_n^i, \theta_n) = x_n^i + \delta_t f_n(x_n^i, \theta_n)$. Similar to equation 3, define the scaled discrete Hamiltonian

$$H_n(x, p, \theta) = p \cdot g_n(x, \theta) - \delta_t L(\theta) \tag{7}$$

In the following algorithms, we employ an augmented variant of Hamiltonian(Li et al., 2018), which additionally subtracts a regularization term of $\frac{1}{2}\rho\|x_{n+1} - g_n(x_n, \theta_n)\|_2^2 + \frac{1}{2}\rho\|p_n - p_{n+1}\nabla_x g_n(x_n, \theta_n)\|_2^2$ from the Hamiltonian discussed in equation 7. Here $\rho$ serves as a hyperparameter, with its reciprocal $1/\rho$ exerting a similar effect as the learning rate.

A modification of the successive approximations method can be employed to address the Pontryagin Maximum Principle (PMP), thereby yielding an alternative training algorithm for deep learning(Li et al., 2018). We present the extended method of successive approximation in Figure 3.

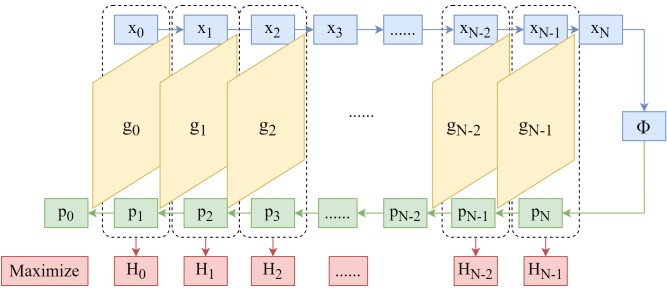

Figure 3: Extended Method of Successive Approximation(E-MSA)

In each iteration, we commence with a feed-forward pass to compute the state $x_i$ for $i = 0, 1, \cdots, N$, followed by a back-propagation step to compute the co-state $p_i$ for $i = N, N-1, \cdots, 0$. Subsequently, we calculate the Hamiltonian for each layer using both the state and co-state, seeking to maximize $H_n$ over $\theta_n$. We iteratively perform these steps to converge towards the desired solution.

## 2.2 L-MSA: Layer-wise Fine-tuning via MSA

As indicated in Section 1, it is often the case that fine-tuning the entire network is unnecessary. Rather, the focus lies in fine-tuning only a single layer or a small subset of layers. In response, we introduce L-MSA, a novel approach for layer-wise fine-tuning that consists of two key components: a metric for layer selection and an optimization algorithm for fine-tuning the selected layer.

We leverage the principles of the Method of Successive Approximations (MSA) (Chernousko & Lyubushin, 1982; Li et al., 2018) as the foundational principle for both layer selection and layer fine-tuning. Our method aims to improve model performance by focusing on the specific strengths of individual layers, targeting the most impactful layers for fine-tuning.

Denote $\Phi(x_N) = \sum_{i=1}^{K} \Phi_i\left(x_N^i\right)$ and $G_n(\cdot) = \Phi \circ g_{N-1}(\cdot, \theta_{N-1}) \circ \cdots \circ g_{n+1}(\cdot, \theta_{n+1})$, which maps the state of the $(n+1)$-th layer to the terminal loss using the latter part of the model. Denote the terminal loss $J = \sum_{i=1}^{K} \Phi_i\left(x_T^i\right)$ as a function of the $(n+1)$-th layer $J_n(\theta_n)$.

**Layer Selection:** A natural approach to layer selection is to choose the layer for fine-tuning that minimizes the loss and thus maximizes performance. In our proposed method, for the layer selection process, we approximate the optimal updated loss resulting from fine-tuning individual layers and use this approximated loss as the criterion for selecting layers.

To approximate the updated loss, we employ the principle of MSA. We begin with a feed-forward pass to compute the state $x_i$ for $i = 0, 1, \cdots, N$, followed by a back-propagation step to compute the co-state $p_i$ for $i = N, N-1, \cdots, 0$. By leveraging both $x_n$ and $p_n$, we approximate the optimal updated loss incurred by fine-tuning individual layers. This process effectively computes the greedy one-step loss for each layer, guiding the fine-tuning to the layer that promises the most immediate improvement in performance. The updated loss after fine-tuning $\theta_n$ can be approximated by

$$\hat{J}_n(\theta_n) = G_{(n+1)}\left(x_{n+1} + \frac{1}{\hat{\rho}_n} p_{n+1} x_n^\top x_n\right) \tag{8}$$

We'll justify this approximation in Section 3. Here $\frac{1}{\hat{\rho}_n}$ acts similarly to the learning rate, and we aim to provide a well-estimated value of the optimal learning rate at an appropriate scale. Notably, the optimal learning rates can vary significantly across different scenarios, even within the same network, where different layers may require distinct values. The accuracy of $\hat{\rho}_n$ plays a crucial role in estimating the updated loss.

Thus, we aim to provide a reasonably accurate estimate of $\hat{\rho}_n$ at the order-of-magnitude level to achieve a precise approximation of the optimal updated loss. We set $\hat{\rho}_n$ as defined in equation 9, computed using the state $x_n$ and co-state $p_n$, with $r_n = p_{n+1} x_n^\top$ and $d'$ being the output dimension. In practice, the terminal loss $J$ may vary in scale. Thus, we sometimes modify $\hat{\rho}_n$ by multiplying it by a constant for all layers.

$$\hat{\rho}_n = \frac{d'}{2J} \cdot \frac{\|p_{n+1}^\top r_n x_n\|_F^2}{\|r_n\|_F^2} \tag{9}$$

We'll demonstrate in Section 3 that it approximates the optimal $\rho_n^*$ in equation 12 to achieve the minimal updated loss within the deep linear network setting. To guide our layer selection process, we utilize $\hat{J}_n(\theta_n)$ in equation 8 as our metric for layer selection, opting to select the layer characterized by the minimal approximated loss. In other words, we select the layer of $g_m(\cdot, \theta_m)$ such that

$$m = \underset{n=0,1,\cdots,N-1}{\arg\min} \hat{J}_n(\theta_n)$$

**Layer Fine-tuning:** Following the layer selection process, we utilize the Method of Successive Approximations (MSA) for fine-tuning the selected layer, with the primary objective of maximizing $H_m$ with respect to $\theta_m$. The MSA process is structured to enhance the optimization of the chosen parameters systematically.

In each iteration, we start with a feed-forward pass through the network to compute the state $x_i$ for each layer, where $i$ ranges from 0 to $N$, capturing the current output based on the input data. Once the state is computed, we proceed to a back-propagation step to derive the co-state $p_i$ for each layer,

starting from the last layer $N$ and moving backward to layer $0$. The co-state represents the sensitivity of the Hamiltonian with respect to the states, providing valuable information for optimization. Next, we compute the Hamiltonian specifically for the layer with $g_m(\cdot, \theta_m)$ using both the state and co-state variables, aiming to maximize $H_m$ over the parameters $\theta_m$ of the selected layer. By repeating these steps iteratively, we progressively refine the layer parameters, converging toward a desired solution that enhances model performance.

Additionally, we have the flexibility to employ alternative optimization algorithms, such as Adam, during this process, which allows us to explore various strategies.

The methodology outlined is visually depicted in Figure 2, offering a comprehensive illustration of the layer-wise fine-tuning process. In Section 3, we will provide a detailed rationale and justification for our chosen metric utilized in the selection of layers.

## 3 THEORETICAL ANALYSIS

In this section, we undertake a theoretical examination of our methodology within the idealized framework of the deep linear network. Given that deep neural networks are composed of linear and activation layers, an analysis of the deep linear network serves as a valuable avenue for gaining insight into our approach. Previous analyses(Arora et al., 2018a;b; Cohen et al., 2023) have provided significant insights into the behavior and properties of deep linear networks, underscoring the importance of this simplified model in understanding more complex architectures.

For simplicity, we employ a simplified variant of the augmented Hamiltonian and consider the maximization step of the $(n + 1)$-th layer as follows:

$$\max_{\theta_n^*} \quad p_{n+1} \cdot g_n(x_n, \theta_n^*) - \frac{1}{2}\rho_n \|\theta_n^* - \theta_n\|_2^2 \tag{10}$$

Given a collection of $K$ sample input-label pairs $\{x^i, y^i\}_{i=1}^K$, with the inputs $x = (x^1, x^2, \cdots, x^K) \in \mathbb{R}^{d \times K}$ and the labels $y = (y^1, y^2, \cdots, y^K) \in \mathbb{R}^{d' \times K}$. Consider an $N$-layer deep linear network

$$x_{n+1} = g_n(x_n, \theta_n) = \theta_n x_n, n = 0, 1, \cdots, N - 1.$$

with the input $x_0 = x$ and the loss function $J = \sum_{i=1}^K \Phi_i(x_N^i) = \frac{1}{2}\sum_{i=1}^K \|y^i - x_N^i\|_2^2$.

**Proposition 3.1** *With given $\rho_n$, the updated loss after fine-tuning $\theta_n$ for one iteration is exactly $\hat{J}_n(\theta_n)$ in Equation 8, i.e.,*

$$J^{update} = G_{(n+1)}\left(x_{n+1} + \frac{1}{\rho_n}p_{n+1}x_n^\top x_n\right) \tag{11}$$

Due to space constraints, the proof details are provided in Appendix A.1.

For simplicity of expression, denote $\beta_n = \theta_{N-1} \cdots \theta_{n+1}$, and $r_n = p_{n+1}x_n^\top = \theta_n^\top \cdots \theta_{N-1}^\top (y - x_N)x_n^\top$ for $n = 0, 1, \cdots, N - 1$. Below we show the relationship between the optimal $\rho_n^*$ and our approximated $\hat{\rho}_n$.

**Proposition 3.2** *The optimal $\rho_n^*$ to achieve the minimal updated loss is*

$$\rho_n^* = \frac{\|\beta_n r_n x_n\|_F^2}{\|r_n\|_F^2} \tag{12}$$

*and it satisfies $\rho_n^* \geq \frac{1}{d'}\hat{\rho}_n$ for the $\hat{\rho}_n$ determined in equation 9.*

*In addition, denote $\hat{\alpha}_n = \frac{1}{\hat{\rho}_n}$ and $\alpha_n^* = \frac{1}{\rho_n^*}$. Let $\theta$ be the 1-dimensional vectorization of all parameters. If $\theta \sim Uniform(B(0, r))$, $\forall r$, i.e., $\theta$ follows a uniform distribution in the neighborhood centered at the origin with radius $r$, we have $E_\theta \alpha_n^* = E_\theta \hat{\alpha}_n$, i.e., we provide an unbiased estimation for $\alpha_n^*$, which functions similarly to a learning rate.*

Due to space constraints, the proof details are provided in Appendix A.1.

# 4 EXPERIMENTAL RESULTS

In this section, we evaluate the performance of our proposed L-MSA method across various datasets. We compare L-MSA against established baseline methods to highlight its effectiveness in selecting optimal layers and improving fine-tuning results. Further details about the datasets and the models are provided in Appendix A.2.

## 4.1 BASELINE METHODS

To compare with other baselines, we follow the setups from prior work(Lee et al., 2022). We employ full fine-tuning as a baseline and focus on the comparison with layer-wise methods such as LIFT(Zhu et al., 2023), LISA(Pan et al., 2024), and surgical fine-tuning(Lee et al., 2022). Among these methods, surgical fine-tuning provides a metric, RGN, for selecting layers. We include a comparison between our proposed metric and theirs to evaluate performance.

**Full Fine-tuning** is a widely used approach for adaptation. The model is initialized with pre-trained weights and biases, and all parameters undergo gradient updates during fine-tuning. In our experiments, we use the Adam optimizer to update all layers of the model.

**LIFT**(Zhu et al., 2023) is a layer-wise method where only one layer(or transform block) is updated in each iteration. The selection policy for updating the layers can follow one of three strategies: (i) front to end, (ii) end to front, or (iii) random. In our experiments, we test all three strategies and report the average performance.

**LISA**(Pan et al., 2024) applies the idea of importance sampling to different layers in LLMs and randomly freezes most middle layers during optimization. LISA consistently fine-tunes the first and last layers, while updating each middle layer with a fixed probability.

**Surgical Fine-tuning**(Lee et al., 2022) shows that selectively fine-tuning a subset of layers matches or outperforms commonly used fine-tuning approaches. The authors propose two criteria for automatically selecting which layers to freeze, with the Relative Gradient Norm (RGN), defined as $RGN = \frac{\|g\|_2}{\|\theta\|_2}$, showing better performance according to their findings. We compare our metric with RGN and also evaluate the performance of our L-MSA method against Auto-RGN, which fine-tunes the layer selected based on the highest RGN value.

## 4.2 EFFECTIVENESS OF OUR METRIC

We first conducted experiments to compute our proposed metric, the approximated optimal updated loss $\hat{J}_n$, and compared it with the true loss after training. In the case of pre-training on ImageNet and fine-tuning on CIFAR-100, represented on the left side of Figure 4, the later layers exhibit smaller approximated updated losses $\hat{J}_n$.

Conversely, when pre-training is done on CIFAR-100 and fine-tuning is applied to a transformed version of the dataset, shown on the right side of Figure 4, the earlier layers show smaller approximated updated losses $\hat{J}_n$. This transformed dataset is created by applying element-wise multiplication to the input data of CIFAR-100 with a matrix, where each entry is $e^x$, and $x$ follows a standard normal distribution. These findings align with the actual training results shown in 1.

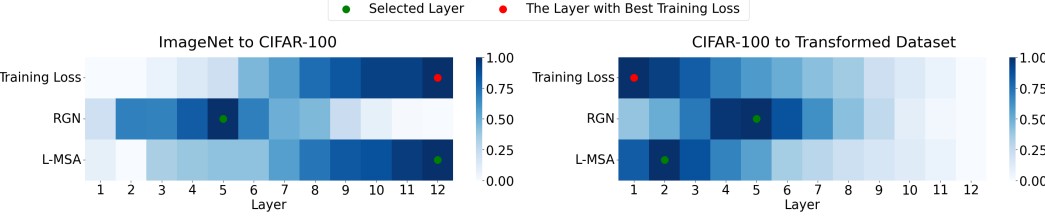

Figure 4: Comparison of our L-MSA metric and RGN with the true training loss. Due to differences in scale, where smaller values are preferred for both our metric and loss while larger values are preferred for RGN, all values are normalized. A darker color indicates a better metric, suggesting that the corresponding layer will be selected for fine-tuning.

We present the comparison of our L-MSA metric and RGN with the true training loss in 4. The results illustrate that our L-MSA metric consistently identifies layers associated with improved training loss, effectively pinpointing those that contribute to better training outcomes. However, in these two cases, RGN assigned the highest metric to the fifth layer, yet it was unable to assist in selecting the more effective layers for fine-tuning.

We also evaluated our metric on four real-data tasks: CIFAR-C, CIFAR-Flip, Living-17, and ImageNet-C. Due to space constraints, the results are provided in Appendix A.3, while the results of fine-tuning the selected layers using our L-MSA method are presented in Section 4.3.

### 4.3 FINE-TUNING RESULTS

We present the results of our L-MSA method in Figure 5, comparing it with Auto-RGN and full fine-tuning for DeiT models fine-tuning from ImageNet to CIFAR-100 and from CIFAR-100 to a transformed dataset. In the case of Auto-RGN, the layer selected by the RGN metric is updated using the Adam optimizer.

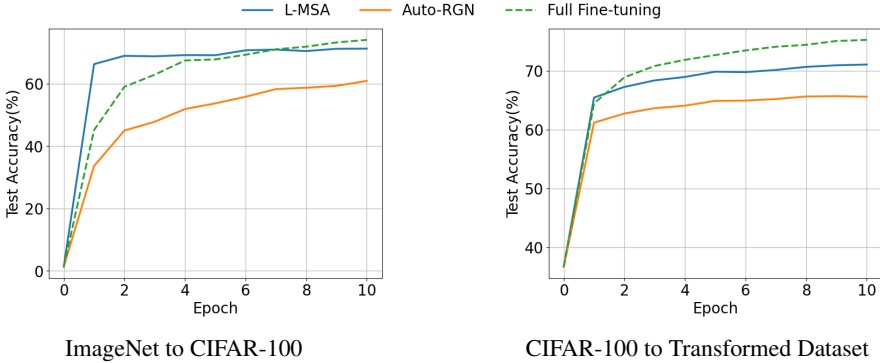

|  ImageNet to CIFAR-100  |  CIFAR-100 to Transformed Dataset  |

Figure 5: The Performance of L-MSA on DeiT-Tiny

In both scenarios, our findings show that the L-MSA method outperforms Auto-RGN and achieves performance comparable to full fine-tuning. Notably, using L-MSA for layer-wise fine-tuning results in performance improvements of up to 20% compared to full fine-tuning and up to 30% compared to Auto-RGN in the initial stages of training. Specifically, we observed a rapid decrease in training loss within the first few batches, underscoring the method's effectiveness, especially in cases where the amount of data is limited.

To further assess the performance of our L-MSA method, we evaluated the performance of our L-MSA method on four real-data tasks with a limited amount of data. For CIFAR-C(Hendrycks & Dietterich, 2019) and CIFAR-Flip(Lee et al., 2022), the models were pre-trained on CIFAR-10(Krizhevsky, 2009) using Wide ResNet-28-10(He et al., 2016). For Living-17(Santurkar et al., 2020) and ImageNet-C(Kar et al., 2022), the models were pre-trained on ImageNet(Deng et al., 2009a) using ResNet-50(He et al., 2016).

|  | CIFAR-C | CIFAR-Flip | Living-17 | ImageNet-C | Average Rank |
|---|---|---|---|---|---|
| No Adaptation | 60.3 | 0.0 | 73.2 | 18.1 | - |
| Full Fine-tuning | 81.1 | 86.2 | 78.2 | **49.0** | 2.5 |
| LIFT | 80.5 | 86.44 | 76.2 | 43.6 | 4.25 |
| LISA | 80.2 | 81.6 | 77.4 | 48.2 | 4.0 |
| Auto-RGN | **82.5** | 88.7 | 77.1 | 48.6 | 2.25 |
| L-MSA | 81.3 | **92.7** | **79.1** | 47.4 | **2.0** |

Table 1: We report the test accuracy on the target distribution across four real-data tasks. Our results show that L-MSA outperforms all other layer-wise fine-tuning methods, including Full Fine-tuning, LISA, LIFT, and Auto-RGN. The best-performing method for each distribution shift is highlighted in bold.

The results, presented in Table 1, compare L-MSA against other fine-tuning approaches, including Full Fine-tuning, LIFT, LISA, and Auto-RGN. Further details on the experimental setup can be found in the Appendix A.2.

The "No Adaptation" baseline provides a reference point for model performance without fine-tuning. L-MSA consistently outperforms other methods, achieving the highest test accuracy on CIFAR-Flip and Living-17, along with the best overall ranking across tasks. Notably, we also observe that L-MSA achieves these results using fewer epochs. Auto-RGN proposed in surgical fine-tuning(Lee et al., 2022) also achieves a competitive average rank.

Overall, L-MSA's strong performance highlights its effectiveness in selecting layers for fine-tuning and utilizing the MSA method to optimize the chosen layer during subsequent fine-tuning. The results emphasize L-MSA's robustness and adaptability, demonstrating its ability to maintain high accuracy across various types of distribution shifts.

### 4.4 EMPIRICAL ANALYSIS

To assess the effectiveness of our proposed L-MSA method, we conducted an ablation study comparing it against other fine-tuning approaches, specifically (i) Full Fine-tuning, (ii) Full fine-tuning using MSA, and (iii) L-MSA Metric + Adam. This comprehensive comparison aimed to evaluate not only the performance of the L-MSA method but also to understand how each approach influences model performance. The average test accuracies across four datasets are plotted in Figure 6.

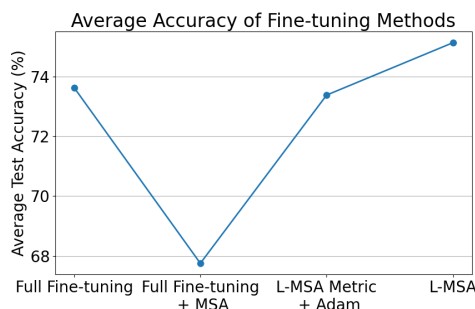

Figure 6: Ablation Study

The results indicate that L-MSA significantly enhances performance compared to other fine-tuning approaches. Notably, the Full Fine-tuning + MSA method underperforms because it optimizes each layer's Hamiltonian individually for multiple steps, which is less effective in the context of full fine-tuning. However, using only the L-MSA metric for layer-wise fine-tuning with Adam achieves performance comparable to that of Full Fine-tuning, demonstrating the metric's effectiveness in layer selection and the advantages of layer-wise fine-tuning. Furthermore, L-MSA outperforms the L-MSA Metric + Adam approach, emphasizing the benefits of utilizing MSA to optimize the selected layers.

## 5 LIMITATIONS AND FUTURE DIRECTIONS

While our layer-wise fine-tuning algorithm shows promising results, it is important to acknowledge its limitations. Firstly, we select layers based on the approximated updated loss, which provides a good estimation of the training loss. However, this does not always guarantee strong generalization to the test data. Additionally, while layer-wise fine-tuning reduces the computational burden compared to full fine-tuning, it may still demand substantial computational resources due to performing both forward and backward propagation, especially in large-scale models.

Future work could explore periodically reselecting layers and adjusting the training configuration after a certain training period, allowing for continuous optimization and more efficient resource use, potentially enhancing performance over time.

## 6 RELATED WORK

### 6.1 LARGE-SCALE MODELS

The emergence of large-scale models has revolutionized various domains, ranging from natural language processing to computer vision. These models, characterized by their extensive parameterization and sophisticated architectures, have demonstrated remarkable capabilities in capturing complex patterns and representations from vast amounts of data.

In natural language processing, models like BERT(Devlin et al., 2018) and GPT(Radford et al., 2018) have set new benchmarks in a variety of tasks, such as language understanding and generation. By leveraging vast text corpora, these models learn rich semantic representations, excelling in various downstream tasks. Similarly, in computer vision, models like ResNet(He et al., 2016) and EfficientNet(Tan & Le, 2019) have demonstrated unprecedented performance in image classification,

object detection, and semantic segmentation tasks. By leveraging large datasets like ImageNet(Deng et al., 2009b), these models learn hierarchical features essential for understanding visual content.

Despite their impressive performance, these models pose significant computational challenges, particularly due to high training costs. Addressing these issues is a key research focus, with ongoing efforts aimed at developing more efficient techniques for both training and inference.

## 6.2 PARAMETER-EFFICIENT FINE-TUNING

Parameter-efficient fine-tuning (PEFT) techniques are designed to adapt pre-trained models by selectively fine-tuning only a subset of parameters. In general, PEFT methods can be categorized into three classes:

**Prompt-based methods** prioritize the optimization of input tokens or input embeddings while keeping the model parameters frozen(Diao et al., 2022; Hambardzumyan et al., 2021; Lester et al., 2021; Liu et al., 2023). Continuous and differentiable forms of prompt engineering (soft prompt) are designed to ease optimization. These approaches typically incur the lowest training cost among the various types mentioned. However, they do not effectively reduce back-propagation costs.

**Adapter methods** typically introduce an auxiliary module with much fewer parameters than the original model. During training, updates are exclusively applied to the adapter module, allowing for more efficient parameter fine-tuning(Diao et al., 2023; Houlsby et al., 2019; Hu et al., 2021). These approaches require manual design and many of them also do not effectively reduce back-propagation costs.

**Selective methods** focus on the optimization of a subset of the model's parameters without the addition of extra modules. For instance, Exclusively fine-tuning bias terms can yield competitive performance comparable to fine-tuning the entire model(Zaken et al., 2021). Recently several noteworthy techniques have been developed in this area, particularly through the concept of layer freezing(Li et al., 2024; Liu et al., 2021). Compared with previous ones, Selective methods are more closely related to our approach.

## 6.3 TRANSFER LEARNING

Previous research in transfer learning has extensively explored the efficacy of fine-tuning to adapt pre-trained features to a target distribution(Oquab et al., 2014; Sharif Razavian et al., 2014; Yosinski et al., 2014). To maintain the valuable information obtained during pre-training, numerous studies have proposed various methods to regularize the fine-tuning process(Li et al., 2020; Shen et al., 2021; Zhang et al., 2020). These methods aim to strike a balance between retaining the learned features from the pre-trained model and adapting to the new target domain, thus ensuring effective knowledge transfer. Notably, several works have demonstrated that freezing certain parameters within the pre-trained model can significantly reduce overfitting during fine-tuning(Kirkpatrick et al., 2017; Lee et al., 2019).

Contrary to most of the prevailing approaches, our work presents a counterintuitive finding: performing fine-tuning on the early layers can yield superior performance in specific scenarios. This intriguing finding resonates with recent investigations in the field(Lee et al., 2022), further undermining the prevailing notion that fine-tuning endeavors should predominantly concentrate on later layers, which are assumed to be more intricately tied to task-specific features.

## 7 CONCLUSION

In conclusion, we have presented L-MSA, a novel layer-wise fine-tuning approach that integrates a metric for layer selection with an optimization algorithm based on the Method of Successive Approximations (MSA). This framework allows for efficient and targeted fine-tuning of individual layers, significantly enhancing model performance while reducing computational costs. Our experiments across various datasets and tasks validate the effectiveness of L-MSA, demonstrating that our method consistently outperforms baseline techniques. By algorithmically determining which layers to fine-tune, we provide a practical solution to the challenges posed by large-scale models. Overall, our work advances the field of layer-wise fine-tuning, offering new insights into optimizing model training and setting the stage for future research in this area.

**Reproducibility Statement:** Source codes for our experiments are provided in the supplementary materials of the paper. The details of our experimental settings and computational infrastructure are given in Section 4 and the Appendix A.2. All datasets that we used in the paper are published, and they are easy to find in the Internet.

**Ethics Statement:** Given the nature of the work, we do not foresee any negative societal and ethical impacts of our work.

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

# Supplement to "L-MSA: Layer-wise Fine-tuning using the Method of Successive Approximations"

**Table of Contents**

## A    APPENDIX

### A.1    THEORETICAL ANALYSIS

In this section, we provide detailed proofs for propositions in Section 3. For simplicity, we consider the case where the batch size is 1. similar results can be derived for a larger batch size.

**Proposition A.1** *With given $\rho_n$, the updated loss after fine-tuning $\theta_n$ for one iteration is exactly $\hat{J}_n(\theta_n)$ in equation 8.*

**Proof:** Given an input-label pair $\{x_0, y\}$, with the inputs $x_0 \in \mathbb{R}^d$ and the labels $y \in \mathbb{R}^{d'}$. Consider an $N$-layer deep linear network

$$x_{n+1} = g_n(x_n, \theta_n) = \theta_n x_n, n = 0, 1, \cdots, N-1.$$

with the input $x_0 = x$ and the loss function $J = \frac{1}{2}\|y - x_N\|_2^2$.

Then we could compute the co-state

$$p_n = \theta_n^\top \cdots \theta_{N-1}^\top (y - x_N), \quad n = 0, 1, \cdots, N.$$

In every iteration, our objective is to maximize the augmented Hamiltonian for a single layer, as shown in equation 10. By taking the derivative of the Hamiltonian with respect to $\theta_n^*$ and setting it to zero, we can derive the updated parameters $\theta_n^*$ as follows.

$$\theta_n^* = \theta_n + \frac{1}{\rho_n} p_{n+1} x_n^\top = \theta_n + \frac{1}{\rho_n} \theta_{n+1}^\top \cdots \theta_{N-1}^\top (y - x_N) x_n^\top \tag{13}$$

which demonstrates that the E-MSA method using the simplified augmented Hamiltonian is equivalent to the gradient descent method with a learning rate of $\alpha_n = \frac{1}{\rho_n}$.

Then we can derive the updated loss as follows.

$$
\begin{aligned}
J^{update} &= \frac{1}{2}\|\beta_n \theta_n^* x_n - y\|_2^2 = G_{(n+1)}\left(\theta_n^* x_n\right) \\
&= G_{(n+1)}\left(x_{n+1} + \frac{1}{\rho_n} p_{n+1} x_n^\top x_n\right)
\end{aligned}
\tag{14}
$$

which is exactly $\hat{J}_n(\theta_n)$ in equation 8. $\qquad\square$

**Proposition A.2** *The optimal $\rho_n^*$ to achieve the minimal updated loss is*

$$\rho_n^* = \frac{\|\beta_n r_n x_n\|_F^2}{\|r_n\|_F^2} \tag{15}$$

*and it satisfies $\rho_n^* \geq \frac{1}{d'}\hat{\rho}_n$ for the $\hat{\rho}_n$ in equation 9.*

*In addition, denote $\hat{\alpha}_n = \frac{1}{\hat{\rho}_n}$ and $\alpha_n^* = \frac{1}{\rho_n^*}$. Let $\theta$ be the 1-dimensional vectorization of all parameters. If $\theta \sim Uniform(B(0, r))$, $\forall r$, i.e., $\theta$ follows a uniform distribution in the neighborhood centered at the origin with radius $r$, we have $E_\theta \alpha_n^* = E_\theta \hat{\alpha}_n$.*

**Proof:** To get the minimum loss that can be achieved after one iteration, we need to compute the optimal $\rho_n$. Let $\beta_n = \theta_{N-1} \cdots \theta_{n+1}$, and $r_n = p_{n+1} x_n^\top = \theta_{n+1}^\top \cdots \theta_{N-1}^\top (y - x_N) x_n^\top$ for $n = 0, 1, \cdots, N - 1$. Thus, we have $p_{n+1} = \beta_n^\top (y - x_N)$ and $\theta_n^* = \theta_n + \alpha r_n$.

$$J^{update} = J_n(\theta_n^*) = \frac{1}{2}\|\beta_n \theta_n^* x_n - y\|_2^2$$
$$= \frac{1}{2}\|\beta_n \theta_n x_n - y + \alpha_n \beta_n r_n x_n\|_2^2 \tag{16}$$

By taking the derivative with respect to $\alpha$ and setting it to zero, we get that the optimal learning rate $\alpha_n^*$ satisfies

$$0 = x_n^\top r_n^\top \beta_n^\top (\beta_n \theta_n x_n - y + \alpha_n^* \beta_n r_n x_n) \tag{17}$$

Therefore, to achieve the minimum loss, we should set

$$\alpha_n^* = \frac{tr(r_n^\top r_n)}{tr(x_n^\top r_n^\top \beta_n^\top \beta_n r_n x_n)} = \frac{\|r_n\|_F^2}{\|\beta_n r_n x_n\|_2^2} \tag{18}$$

where $\|\cdot\|_F$ is the Frobenius norm. Thus, we can derive that the optimal $\rho_n^*$

$$\rho_n^* = \frac{1}{\alpha_n^*} = \frac{\|\beta_n r_n x_n\|_2^2}{\|r_n\|_F^2} \geq \frac{\|(y - x_N)^\top \beta_n r_n x_n\|_2^2}{\|(y - x_N)\|_2^2 \cdot \|r_n\|_F^2} = \frac{1}{2J} \cdot \frac{\|p_{n+1}^\top r_n x_n\|_F^2}{\|r_n\|_F^2} = \frac{1}{d'}\hat{\rho}_n \tag{19}$$

Consider

$$\gamma = \frac{\|(y - x_N)^\top \beta_n r_n x_n\|_2^2}{\|y - x_N\|_2^2 \|\beta_n r_n x_n\|_2^2} = \frac{\|(y - x_N)^\top \beta_n \beta_n^\top (y - x_N) x_n^\top x_n\|_2^2}{\|y - x_N\|_2^2 \|\beta_n \beta_n^\top (y - x_N) x_n^\top x_n\|_2^2}$$
$$= \frac{\|(y - x_N)^\top \beta_n \beta_n^\top (y - x_N)\|_2^2}{\|y - x_N\|_2^2 \|\beta_n \beta_n^\top (y - x_N)\|_2^2} \tag{20}$$

To evaluate $\gamma$, without loss of generality, we can set $\|y - x_N\|_2 = 1$. There exists an orthogonal matrix $Q \in \mathbb{R}^{d' \times d'}$ such that $y - x_N = Q \cdot e_1$, where $e_1 = (1, 0, \cdots, 0)^\top \in \mathbb{R}^{d'}$. We denote $Q^\top \beta_n \beta_n^\top Q = (b_{ij})_{1 \leq i,j \leq d'}$. Then

$$\gamma = \frac{\|e_1^\top Q^\top \beta_n \beta_n^\top Q e_1\|_2^2}{\|\beta_n \beta_n^\top Q e_1\|_2^2} = \frac{\|e_1^\top Q^\top \beta_n \beta_n^\top Q e_1\|_2^2}{\|Q^\top \beta_n \beta_n^\top Q e_1\|_2^2}$$
$$= \frac{b_{11}^2}{b_{11}^2 + b_{21}^2 + \cdots + b_{d'1}^2} \tag{21}$$

$\forall r$, if $\theta \sim Uniform(B(0, r))$, i.e., $\theta$ follows a uniform distribution in the neighborhood centered at the origin with radius $r$, we have $E_\theta \gamma = \frac{1}{d'}$. As for the expectation of $\alpha^*$, we have

$$\alpha_n^* = \frac{\|r_n\|_F^2}{\|\beta_n r_n x_n\|_2^2} = \gamma \cdot \frac{\|(y - x_N)\|_2^2 \cdot \|r_n\|_F^2}{\|(y - x_N)^\top \beta_n r_n x_n\|_2^2}$$
$$= \gamma \cdot 2J \cdot \frac{\|r_n\|_F^2}{\|p_{n+1}^\top r_n x_n\|_F^2} \tag{22}$$

According to our definition

$$\hat{\alpha}_n = \frac{1}{\hat{\rho}_n} = \frac{2J}{d'} \cdot \frac{\|r_n\|_F^2}{\|p_{n+1}^T r_n x_n\|_F^2} \tag{23}$$

Therefore, we can get the result that

$$E_\theta \alpha_n^* = E_\theta \hat{\alpha}_n$$

$\square$

## A.2 EXPERIMENTAL SETUP

In this section, we elaborate on the experimental setup used in our study, covering various aspects such as datasets, models, pretraining, training details, and environment.

### A.2.1 DATASETS

**CIFAR-100**(Krizhevsky, 2009): A widely used benchmark dataset consisting of 100 classes, with 600 images per class. The dataset is split into 50,000 training images and 10,000 test images, with each image being a 32x32 RGB image.

**CIFAR-10-C**(Hendrycks & Dietterich, 2019): A corrupted version of CIFAR-10, containing 10 classes with various common corruptions applied to the original test set, making it ideal for evaluating model robustness under real-world noisy conditions. The dataset has 5 levels of severity, and we evaluate with the most severe level. We tune on 1000 images from CIFAR-10-C.

**CIFAR-Flip**(Lee et al., 2022): A synthetic task where the inputs are from the original CIFAR-10 dataset, but the target labels are flipped such that each label $y$ is transformed to $9 - y$ (e.g., label $0$ becomes label $9$, label $1$ becomes label $8$, etc.). This task provides a controlled setting to assess model performance when faced with adversarially shifted labels.

**Living-17**(Santurkar et al., 2020): A specialized dataset for classifying living organisms, including animals and plants, with 17 distinct classes. It presents a challenging test case due to its domain-specific nature, requiring the model to differentiate between fine-grained categories of organisms. For Living-17, we tune on 850 images from the target distribution, evenly split between the 17 classes, giving 50 images per class.

**ImageNet-C**(Kar et al., 2022): A corrupted version of the ImageNet dataset, where common corruptions such as Gaussian noise, blur, and weather distortions have been applied to the validation set. Similar to CIFAR-10-C, the dataset has 5 levels of severity, and we evaluate with the most severe level. We tune on 5000 images from ImageNet-C, evenly split between classes, giving 5 corrupted images per class.

### A.2.2 MODELS

Here presents the models utilized in our experiments, highlighting their architectural characteristics and parameter counts. We examine the Tiny version of DeiT(Touvron et al., 2021), Wide ResNet-28-10, and ResNet-50(He et al., 2016), each selected for its unique strengths in handling various image classification tasks. The number of parameters for each model is given in Table 2.

| Model | Number of Parameters |
|---|---|
| DeiT (Tiny) | 5.7 million |
| Wide ResNet-28-10 | 36 million |
| ResNet-50 | 25.6 million |

Table 2: Number of Parameters for Various Models

**DeiT (Data-efficient Image Transformers - Tiny)**(Touvron et al., 2021): The Tiny version of DeiT is a transformer-based model designed for image classification. It utilizes attention mechanisms to capture long-range dependencies in images and achieves high performance with less data through efficient training strategies. In our experiments, it is used on the CIFAR-100 dataset for its ability to generalize well across diverse visual features.

**Wide ResNet-28-10**(He et al., 2016): Wide ResNet-28-10 is a convolutional neural network (CNN) characterized by its wider architecture, which incorporates residual connections to alleviate the vanishing gradient problem. With 28 layers and a width factor of 10, it balances capacity and efficiency, making it suitable for various image classification tasks. We apply Wide ResNet-28-10 to the CIFAR-10-C and CIFAR-Flip datasets to assess its performance under corrupted and adversarially shifted labels.

**ResNet-50**(He et al., 2016): ResNet-50 is a deeper CNN with 50 layers, providing greater capacity for learning complex patterns. It is well-regarded for its robustness in both general and domain-specific challenges. In our study, ResNet-50 is employed on the Living-17 and ImageNet-C datasets, leveraging its strong feature extraction capabilities against corrupted data.

### A.2.3 PRETRAINING

For the CIFAR-100 dataset, we utilize a DeiT-Tiny model pre-trained on ImageNet. For the CIFAR-10-C and CIFAR-Flip datasets, we employ Wide ResNet-28-10 models that have been pre-trained on CIFAR-10. For the ImageNet-C dataset, we use a ResNet-50 model pre-trained on ImageNet.

For the Living-17 dataset, we load a ResNet model pre-trained on ImageNet and train it on the source data for 5 epochs. Initially, we tune only the head for 3 epochs, followed by fine-tuning all layers for an additional 2 epochs, following the Linear Probing then Fine-tuning(LP-FT) strategy(Kumar et al., 2022), while utilizing the Adam optimizer.

### A.2.4 TRAINING DETAILS

**Layer Configuration**: For the DeiT model, each transformer block is treated as a layer, resulting in a total of 12 layers. In the case of Wide ResNet-28-10 and ResNet-50, each block is considered a layer, with the input convolutional layer (conv1) combined into the first layer and the output fully connected layer treated as a separate layer. Consequently, Wide ResNet-28-10 consists of 3 blocks plus the output layer, totaling 4 layers, while ResNet-50 comprises 4 blocks and the output layer, resulting in 5 layers.

**Fine-Tuning with Adam Optimizer**: For all baseline methods, we fine-tune the model on the labeled target data for a total of 10 epochs, with a batch size of 64. We explore 3 different learning rates: $1 \times 10^{-3}$, $1 \times 10^{-4}$, and $1 \times 10^{-5}$ for all methods, except that for CIFAR-Flip, we adjust the learning rates to $1 \times 10^{-1}$, $1 \times 10^{-2}$, and $1 \times 10^{-3}$ for last-layer fine-tuning.

**Fine-Tuning with L-MSA**: We fine-tune the model on the labeled target data for 10 epochs for CIFAR-100 and 5 epochs for other datasets, using a batch size of 64 and setting the hyper-parameter $\rho$ to 1. To optimize the maximization Hamiltonian, we utilize the Adam optimizer with a learning rate of $1 \times 10^{-4}$. The optimization process runs for a total of 5 iterations.

**Validation Stategy**: For all datasets and experiments, we implement early stopping based on the accuracy observed on a held-out validation subset of the labeled target data. We report the test accuracy corresponding to the epoch with the highest validation accuracy.

### A.2.5 ENVIRONMENT

The experiments were conducted on a machine equipped with an NVIDIA GeForce RTX 3090 GPU, which has 24GB of memory.

By providing comprehensive details of our experimental setup, we aim to facilitate transparency and reproducibility in our study, enabling other researchers to validate and build upon our findings accurately.

### A.3 ADDITIONAL EXPERIMENTAL RESULTS

We present the comparison of our L-MSA metric and RGN with the true training loss on four real-world tasks: CIFAR-C, CIFAR-Flip, Living-17, and ImageNet-C. This comparison, omitted from Section 4.2 due to space limitations, is now provided here. As noted previously, due to differences in scale, where smaller values are preferred for both our metric and loss while larger values are preferred for RGN, all values are normalized. A darker color indicates a better metric, suggesting that the corresponding layer will be selected for fine-tuning.

Figure 7 illustrates that our L-MSA metric consistently identifies layers associated with improved training loss, successfully highlighting those that enhance training outcomes. In contrast, RGN often selects layers that do not match the optimal ones, making it less effective in comparison.

However, it is important to emphasize that our layer selection is primarily based on the approximated updated loss, which provides a solid estimate of the training loss. While this metric offers valuable

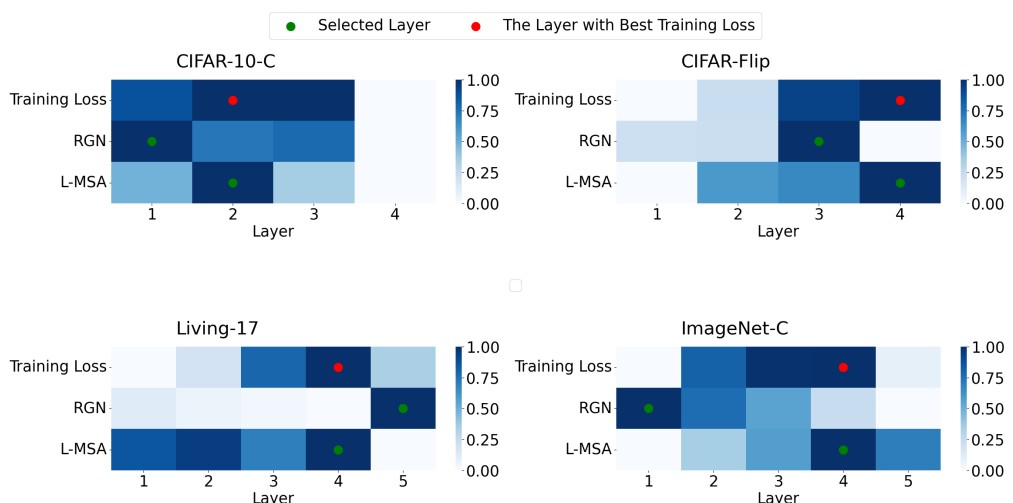

Figure 7: Comparison of L-MSA metric and RGN with the true training loss

insights for identifying effective layers, it is important to recognize that such an approach does not always guarantee robust generalization to the test data.

