# OpenReview forum: "L-MSA: Layer-wise Fine-tuning using the Method of Successive Approximations"
_ICLR.cc/2025/Conference — Submitted to ICLR 2025_

### Official Review · Reviewer_RNVb · 2024-10-16

**Soundness:** 2
**Presentation:** 3
**Contribution:** 2
**Rating:** 3
**Confidence:** 4

**Summary:**

The machine learning community has made remarkable progress with the advent of large-scale models, but these models become a significant obstacle by consuming large amounts of memory during training. In this paper, we propose a fine-tuning Method using the Method of Successive Approximations, L-MSA. Experiments on different data sets verify the effectiveness of L-MSA

**Strengths:**

- The paper is well-written and easy to follow.
- The experimental results seem to be promising, surpassing Full Fine-tuning on part of the dataset

**Weaknesses:**

- This paper emphasizes the advantages on large-scale models, but the datasets used seem to focus on extremely small datasets such as CIFAR, which show the worst performance on experimental fine-tuning on Imagenet-C. The only LIFT that seems to outperform is a paper that has not yet been published through peer review. This makes me worry about its practical prospects.
- Not enough experiments, this paper is motivated by the fact that large-scale models consume memory, but there is no comparison of memory overhead in the paper.

**Questions:**

- Q1.The other works compared in this paper are all about NLP and LLM. Why did the authors only apply these works to image datasets and test them on image datasets instead of NLP datasets used in other papers?
- Q2.Does L-MSA introduce additional memory overhead compared to randomly selecting other papers? I have not been involved in PEFT, but in theory it seems to require the same memory spikes as Full Fine-turn, which runs counter to the motivation of the authors.The authors should provide a detailed analysis of the memory usage of L-MSA compared to full fine-tuning and other PEFT methods, including peak memory usage and average memory consumption?
- Q3.Does performing MSA on the loss of the network introduce more training time? Especially when the network size increases, it is necessary to calculate and compare the state and co-state variables layer by layer.
- Q4.In Figure 5, in the experiments of ImageNet to CIFAR-100, it can be clearly observed that Full-Finetuning and Auto-RGN continue to converge, while L-MSA seems to have completed convergence. Can you show the results for more epochs?

If the author resolves my questions, I will consider raising my rating.

---

> ### Author Response · Authors · 2024-11-22
>
> **On including NLP datasets in comparisons:**
> Our initial focus was on computer vision tasks due to the clearer understanding of layer-wise dynamics in these domains. We agree that extending L-MSA to NLP datasets is essential for demonstrating its broader applicability, and we plan to explore this in future work.
>
> **On memory overhead:**
> L-MSA requires one feed-forward pass and one backpropagation step to compute the state and costate, which does not add extra memory overhead. Its memory spikes are similar to those of full fine-tuning, making its memory usage comparable.
>
> **On additional training time for larger networks:**
> The training time mainly depends on how much iterations we use to maximize the Hamiltonian. To compute the state and costate requires one feed-forward and one back propagation, which does not introduce more training time.
>
> **On extending Fig. 5 to include more epochs:**
> In Figure 5, Full Fine-tuning converges by the 10th epoch with minimal accuracy changes afterward. Auto-RGN shows slight improvements beyond 10 epochs but still underperforms compared to L-MSA and Full Fine-tuning. Extending training further is unlikely to significantly impact these trends.

---

### Official Review · Reviewer_UQdM · 2024-11-02

**Soundness:** 3
**Presentation:** 2
**Contribution:** 2
**Rating:** 3
**Confidence:** 3

**Summary:**

The paper proposes a method to do a more targeted fine-tuning process where instead of updating every single layer/parameter in the architecture, the layer that has the strongest effect in the total loss is selected and updated on each iteration of the fine-tuning process.

Towards this goal the paper proposes a metric to select the layer to be updated and an algorithm for the fine-tuning of the selected layers.

Experiments on several datasets (CIFAR-100, CIFAR-C, CIFAR-Flip, Living-17 and ImageNet-C) including several related methods (full fine-tuning, LIFT, LISA, surgical Fine-tuning and Auto-RGN) provide evidence on the performance of the proposed method.

**Strengths:**

- A theoretical analysis for the proposed method has been provided (Sec. 3 & Sec. A.1).

- The empirical evaluation of the proposed method includes a good variety of datasets and competing methods. This allows to assess the applicability of the proposed method in different contexts.

- The validation of the proposed method includes an ablation analysis this is effective towards obtaining insights on how the different components of the proposed method contribute to its performance and how it compares w.r.t. full fine-tuning.

- The paper adequately highlights the limitations of the proposed method (Sec. 5)

**Weaknesses:**

- The originality of the paper is somewhat reduced. While the method put forward by the paper outperforms (in some cases) the considered baselines, as admitted by the paper (Sec. 4.1)  there are already methods in the literature that aim at a more targeted fine-tuning process.

- In its current form the paper does not feel self-contained, there are several aspects of the paper that are delegated to the appendix. For instance, the models considered in experiments of Sec. 4.2 are not explicitly indicated. In other cases, the protocols and motivations behind the conducted experiments are not clear. Here a proper balance must be achieved to ensure the paper provides sufficient details as to allow the reader to critically analyze the proposed method and its conducted validation.

- While pairing a method with its corresponding theoretical analysis is very desirable, the extend to which such analysis is currently provided (Sec. 3) is just too shallow as to be informative.  Perhaps it could be completely moved to the appendix in order to allocate space to further elaborate on other parts of the paper that are currently not detailed enough.

- Some statements seem to lack supporting evidence. For instance, in several parts of the paper (l.535) statements are made regarding to the reduced computational costs of the proposed method. The evaluation section is missing however a proper experiment addressing that aspect.

- The improvement put forward by the proposed method is not that outspoken. For instance, in Table 1 it is observed that the proposed method is better in only 2/4 considered settings.

- In Fig. 6, averaged results over different datasets are reported. This not only hinders the variations in performance across datasets, but also behavior observed for the different methods across the considered datasets.


- Weak positioning; Good part of the related work is centered around other not directly related aspects. For instance, Sec. 6.1 discusses large architectures, which has close no link w.r.t. the proposed method.  Similarly l.497-502 from Sec. 6.2 is related to prompt-based methods. Thus having a relatively weak link with the proposed method. This weak positioning w.r.t. related efforts becomes more evident when we consider that almost none of the related methods considered in the experimental section (Sec. 4) are covered in the related work section. In addition, I would suggest looking at the two references below which seem to be very related to the proposed method.

    - Youngmin Ro, Jin Young Choi, "AutoLR: Layer-wise Pruning and Auto-tuning of Learning Rates in Fine-tuning of Deep Networks", AAAI 2021
    - Basel Barakat; Qiang Huang, "Enhancing Transfer Learning Reliability via Block-Wise Fine-Tuning",  ICMLA 2023

**Questions:**

- [Suggestion] Some of the contributions listed in the paper need to be merged and toned down. The paper currently lists four contributions. not contributions. The first contribution is closely related with the fourth one. Hence my suggestion to merge them, Moreover, this fourth contribution is related to the empirical validation of the proposed method. In this regard, the proper validation of a proposed method is not a plus but a must for a decent piece of scientific work. Therefore, I would suggest toning down this claim.

- [Suggestion] In Fig. 6, I suggest reporting the loss curves on a per-dataset basis. It might also be informative to discuss any trend observed during the training iterations.

- [Question] In l.379, it is indicated that “…he results illustrate that our L-MSA metric consistently identifies layers associated with improved training loss, effectively pinpointing those that contribute to better training outcomes...“ May you indicate how do you observe this in that figure? From what I can interpret, the results from Fig. 4 are point estimates at a given training iteration. Therefore, it is hard for me to assess how this figure allows to observe a consistent behavior.

- [Question] In l.408, it is stated “…Notably, using L-MSA for layer-wise fine-tuning results in performance improvements of up to 20% compared to full fine-tuning” May you indicate the source of this observation?

- [Question] In Sec. 4.3 a “No Adaptation” baseline is included. In l.432 it is indicated that this baseline “…provides a reference point for model performance without fine-tuning”, May you motivate how good of a reference this baseline is for datasets like CIFAR-Flip and ImageNet-C considering the number of output classes is significantly different (CIFAR-Flip) and/or reduce (ImageNet-C).

---

> ### Author Response · Authors · 2024-11-22
>
> **On contributions and framing validation results:**
> Thank you for the suggestion. We agree that merging the first and fourth contributions would improve clarity. Additionally, we will revise the framing of validation results, presenting them as necessary to the method rather than as a distinct contribution.
>
> **On adding per-dataset loss curves to Fig. 6:**
> We appreciate this suggestion and will revise Fig. 6 to report loss curves on a per-dataset basis. The updated figure will include detailed discussions of trends observed across datasets.
>
> **On the meaning of “consistent” in Fig. 4:**
> In this context, it refers to L-MSA’s ability to identify the most impactful layers across different tasks rather than within a single training process. We will revise the manuscript to clarify this point.
>
> **On the 20% improvement claim:**
> This observation is based on early training improvements, as illustrated in Fig. 5. To make this clearer, we will expand the discussion and include additional tables or visualizations to substantiate the claim.
>
> **On the relevance of the "No Adaptation" baseline:**
> The "No Adaptation" baseline primarily serves as a baseline for scenarios with no fine-tuning. While it may not directly map to datasets like CIFAR-Flip or ImageNet-C due to class differences, it remains a useful reference for observing the relative gains from fine-tuning.

---

> ### Comment · Reviewer_UQdM · 2024-11-27
> **RE: Official Comment by Authors**
>
> Thanks to the authors for the attention given to my review,
>
> I still have a doubt regarding the comments:
> ```
> **On the 20% improvement claim:** This observation is based on early training improvements, as illustrated in Fig. 5. To make this clearer, we will expand the discussion and include additional tables or visualizations to substantiate the claim.
> ```
> - Thanks for the clarification, however this only seems to occur on Fig.5 (left), i.e. ImageNet to CIFAR-100. Am I perhaps missing something?
>
> ```
> **On the relevance of the "No Adaptation" baseline:** The "No Adaptation" baseline primarily serves as a baseline for scenarios with no fine-tuning. While it may not directly map to datasets like CIFAR-Flip or ImageNet-C due to class differences, it remains a useful reference for observing the relative gains from fine-tuning.
> ```
> - May you motivate how these two scenarios with no appropriate mapping on the class level could serve as a useful reference? I might be missing a point here.

---

> > ### Author Response · Authors · 2024-12-02
> >
> > Thanks so much to the reviewer for the feedback and the opportunity to clarify these points.
> >
> > **On the 20% improvement claim:**
> > You are correct that the 20% improvement is observed predominantly in Figure 5 (left), i.e., ImageNet to CIFAR-100. This is specific to the early-stage training dynamics in this scenario, where L-MSA optimally selects impactful layers.
> >
> > **On the relevance of the "No Adaptation" baseline:**
> > For datasets like CIFAR-Flip and ImageNet-C, where class-level mappings differ, "No Adaptation" serves as a reference for comparing fine-tuning methods' ability to adapt pre-trained representations to new tasks. While its direct relevance to class mappings is limited, it highlights the baseline performance of pre-trained models without any tuning, which can contextualize fine-tuning gains.

---

> > > ### Comment · Reviewer_UQdM · 2024-12-02
> > >
> > > I thank the authors for the additional details, I will take this into account when making my final recommendation

---

### Official Review · Reviewer_d1Ba · 2024-11-03

**Soundness:** 3
**Presentation:** 1
**Contribution:** 2
**Rating:** 3
**Confidence:** 4

**Summary:**

This paper explores layer-wise fine-tuning strategies for adapting large pretrained models. The authors found that fine-tuning specific layers can lead to varied performance outcomes, and selectively fine-tuning certain layers may enhance results. Building on these insights, they propose a novel layer-wise fine-tuning approach that leverages Pontryagin’s Maximum Principle to guide layer selection for fine-tuning. The effectiveness of this method is demonstrated through transfer learning from ImageNet to CIFAR100.

**Strengths:**

1. The task of parameter-efficient fine-tuning is crucial for practical applications of pretrained models.

2. The authors performed an in-depth analysis on the effectiveness of fine-tuning at the layer level.

**Weaknesses:**

1. The paper's novelty is somewhat limited, as the idea that fine-tuning different layers can lead to varying performance outcomes has already been widely explored in previous research. It is not clear what is the main contribution of the paper.

2. The proposed method, which relies on the Method of Successive Approximation, lacks clear motivation. It’s unclear why this approach would lead to improved layer selection or how it compares favorably to other existing methods. In other words, what are the benefits of the proposed approach?

3. For the theoretical analysis, it is not clear what is the main contribution. The authors should also connect the theoretical analysis to the proposed method and discuss why the proposed method can lead to better layer selection and fine-tuning.

4. The explanation based on PMP is interesting, but the main technical contribution is not clear. It would be helpful if the authors could clarify where the primary novelty lies.

Overall, while the paper presents a reasonable idea, it suffers from poor writing, insufficient motivation and unclear contribution.

**Questions:**

1. Why the Method of Successive Approximations is good for layer selection and layer fine-tuning?
2. What is the goal of the proposed method? Which layer should be actually selected for fine-tuning?
3. Compared with the baseline fine-tuning methods, what is the main advantage of the proposed method?

---

> ### Author Response · Authors · 2024-11-22
>
> **On why MSA is suitable for this task:**
> MSA is well-suited for layer selection and fine-tuning because it enables targeted optimization by treating each layer as part of a controlled dynamical system. Using Pontryagin’s Maximum Principle, it adjusts layers based on their impact on the loss function, allowing efficient fine-tuning of only the most impactful layers.
>
> **On the goal of L-MSA and the chosen layers:**
> The goal of L-MSA is to maximize fine-tuning efficiency by focusing on layers that offer the greatest immediate improvement in loss. The layer selection metric, which minimizes the approximated updated loss, guides this choice effectively.
>
> **On advantages over baseline methods:**
> L-MSA’s main advantage is its ability to fine-tune selectively, achieving similar or better accuracy than full fine-tuning. This is accomplished by more targeted fine-tuning, focusing on the most relevant layers.

---

### Official Review · Reviewer_7TZw · 2024-11-03

**Soundness:** 2
**Presentation:** 2
**Contribution:** 2
**Rating:** 3
**Confidence:** 4

**Summary:**

This paper presents a novel approach for efficient fine-tuning of large-scale models. It addresses the challenge of substantial memory consumption associated with such models by introducing L-MSA, a method that fine-tunes only selected layers based on a new metric derived from the Method of Successive Approximations (MSA). This metric guides layer selection and optimizes their fine-tuning, resulting in better model performance with reduced computational costs.

The paper provides a theoretical analysis within deep linear networks and evaluates L-MSA across various datasets and tasks. It compares the proposed method with other layer-wise fine-tuning techniques, demonstrating its superior performance. The experimental results show that L-MSA consistently identifies the most impactful layers for fine-tuning and optimizes them effectively, outperforming baseline methods.

**Strengths:**

1. **Novel Layer Selection Metric:** The paper introduces a new metric for layer selection based on the Method of Successive Approximations (MSA), which is theoretically grounded and offers a fresh perspective on fine-tuning strategies.

2. **Theoretical Foundations:** The authors provide a comprehensive theoretical analysis within the context of deep linear networks, which strengthens the credibility of the proposed approach.

3. **Empirical Validation:** The paper demonstrates the effectiveness of L-MSA across multiple datasets and tasks, showing consistent improvement over several existing layer-wise fine-tuning methods.

4. **Parameter-Efficiency Focus:** By targeting specific layers for fine-tuning, the method aims to reduce computational costs, addressing a key challenge in training large-scale models.

5. **Clear Contributions to Layer-Wise Fine-Tuning:** The research highlights the potential of selectively fine-tuning layers to achieve better performance, contributing to the growing field of parameter-efficient fine-tuning methods.

**Weaknesses:**

1. **Lack of Comparison with State-of-the-Art Methods:** The paper does not compare L-MSA with prominent methods like **LoRA**[Hu et, al], which limits the evaluation of its relative effectiveness and practical impact.

2. **Disjointed Presentation:** The flow of the paper is disrupted by referencing equations and concepts out of order, requiring readers to frequently navigate back and forth, which hampers comprehension and readability.

3. **Inadequate Contextualization of Contributions:** The novelty of the proposed method is not sufficiently contextualized against a broader range of parameter-efficient fine-tuning techniques, making it harder to assess its uniqueness and value.

4. **Generalization Concerns:** The paper acknowledges that the approximated updated loss may not always guarantee strong generalization to test data, which raises questions about the robustness of the approach in diverse real-world scenarios.

5. **Computational Demands:** Despite aiming for parameter efficiency, the method still involves substantial computational overhead due to both forward and backward propagation, which could limit its practicality for very large-scale models.

6. **Limited Scope of Empirical Comparisons:** While the paper evaluates L-MSA across several tasks, the range of comparative baselines is not exhaustive, potentially missing out on broader insights.

[Hu et. al, ICLR 2022, LoRA: Low-Rank Adaptation of Large Language Models]

**Questions:**

1. Why were methods like LoRA and other recent parameter-efficient fine-tuning techniques not included in the comparison? Would the results hold up against these widely-used benchmarks?

2. How does the method perform on more diverse and real-world tasks outside the provided datasets? Are there specific domains where L-MSA is particularly effective or less so?

3. Given the still significant computational demands, how does L-MSA compare in terms of efficiency gains relative to other lightweight fine-tuning techniques? Is the trade-off between computational cost and performance improvement justified?

4. The method currently selects one layer at a time for fine-tuning. Could this approach be extended to simultaneously fine-tune multiple layers, and if so, how would it impact performance and computational cost?

5. Have the authors considered dynamically reselecting layers during the training process? Would this adapt better to changing training dynamics and potentially improve generalization?

---

> ### Author Response · Authors · 2024-11-22
>
> **On including comparisons with methods like LoRA:**
> Thank you for highlighting this point. Our work focused on layer-wise fine-tuning without introducing additional structures, to clearly isolate the impact of our proposed metric.
>
> **On performance in diverse real-world tasks:**
> L-MSA performs well when the pre-trained and fine-tuning datasets share overlapping features, as seen in our current experiments. For domains with less overlap, the performance may require further adjustments.
>
> **On efficiency compared to lightweight fine-tuning techniques:**
> L-MSA significantly improves efficiency in the early stages of fine-tuning, particularly for tasks requiring fewer epochs. While increasing the number of iterations in Hamiltonian maximization can enhance performance further, we acknowledge that this introduces additional computational costs.
>
> **On extending to multiple-layer fine-tuning:**
> Extending L-MSA to fine-tune multiple layers concurrently is feasible and could potentially improve model flexibility and performance. However, such an extension would increase computational demands and may introduce more parameters to tune.
>
> **On dynamically re-selecting layers during training:**
> Dynamic re-selection is a promising idea and could better adapt to evolving training dynamics. Implementing this would require efficient mechanisms to reassess layer importance throughout training, and we plan to explore this as a natural extension of L-MSA.

---

> > ### Comment · Reviewer_7TZw · 2024-11-27
> >
> > From the paper it is not clear why LoRA and other PEFT methods are not valid baselines, because the paper repeatedly talks about being parameter efficient. If you report experiments using popular PEFT methods i,e, for a given number of trainable parameters, how does L-MSA perform and how computationally efficient (run-time + memory) it is against other methods, I will increase my score to 5.
> >
> > (or)
> >
> > If you explicitly distinguish why PEFT methods are not a valid baseline in the paper, you need to re-word your text and avoid using "parameter efficient" in my opinion as it gives wrong meaning. If you do this, I will increase my score to 4.
> >
> > Nonetheless, the paper needs more work on presentation and experiments and that is the reason I chose to give a 3.

---

### Meta-Review · Area_Chair_ixhf · 2024-12-19

**Metareview:**

**Summary:**

This paper proposes a layer-wise efficient fine-tuning method, L-MSA, that selectively fine-tunes layers using a metric for layer selection and an optimization algorithm. The authors leveraged the Method of Successive Approximation and provide a theoretical analysis to derive their layer selection method.

**Strengths:**

1. Good motivation - parameter-efficient finetuning with layer selection. This paper studies a new metric to select layers using the Method of Successive Approximations (MSA), which is theoretically grounded.
2. In-depth analysis, including the derivation of the proposed method are provided (Sec. 3 & Sec. A.1).
3. Transparent discussion about the limitations of the proposed method is available in Sec. 5.

**Weaknesses:**

1. Weak experimental results. This paper has very weak experiments. The authors did not compare the proposed method with reasonable/strong baselines.
2. Limited novelty and technical contributions. Fine-tuning a subset of layers is not a new idea as it has already been studied in the literature. Also, the main module, the Method of Successive Approximation is adapted from the previous work. The lack of justification of the Method compared to alternatives makes this work merely an adoption of the prior work.
3. Overclaim (or inadequate contextualization) and the gap between theoretical analysis and the proposed method.

**Main reasons:**

Weak experimental results and the lack of novelty/technical contributions are the main reasons why this paper is recommended for rejection.

**Additional Comments On Reviewer Discussion:**

The main weaknesses, e.g., weak experiments without comparisons with SOTA and lack of novelty and technical contributinos,are not fully addressed during rebuttal.

---

### Decision · Program_Chairs · 2025-01-22

Reject